# Increased Risk of Local Recurrence in Cutaneous Squamous Cell Carcinoma Arising in Sun-Exposed Skin: A Retrospective Cohort Study

**DOI:** 10.3390/cancers15205037

**Published:** 2023-10-18

**Authors:** Ana Morelló-Vicente, Marta Espejo-Marín, Inés Oteiza-Rius, Javier Antoñanzas, Antonio Vélez, Rafael Salido-Vallejo

**Affiliations:** 1Dermatology Department, University Clinic of Navarra, 31008 Pamplona, Spain; amorellovic@unav.es (A.M.-V.); ioteiza@unav.es (I.O.-R.); jantonanzas@unav.es (J.A.); 2Occidente-Azahara Health Care Center, 14005 Córdoba, Spain; marta.espejo.marin.sspa@juntadeandalucia.es; 3Dermatology Department, Reina Sofía University Hospital, 14004 Córdoba, Spain; antonioj.velez.sspa@juntadeandalucia.es

**Keywords:** cutaneous squamous cell carcinoma, local recurrence, metastases, sun exposure

## Abstract

**Simple Summary:**

The aim of this study is to compare the risk of local recurrence and metastases in patients with cSCC based on the presence or absence of prior sun exposure in the region of tumor development. A retrospective observational epidemiological study including 558 patients from January 2017 to December 2020 was conducted. Among the 463 patients with cSCC in highly sun-exposed areas, 73 (15.8%) were diagnosed with local recurrence versus only 7 of 95 patients (7.4%) in less sun-exposed areas. No differences were found in terms of metastasis between both groups. In regions with low sun exposure, the variables linked to a heightened risk of recurrence include tumor depth and the involvement of surgical margins. Our results suggest that highly sun-exposed areas could have a greater risk of developing local recurrence, conferring a worse prognosis for the patients.

**Abstract:**

Background: The incidence of cutaneous squamous cell carcinoma (cSCC) is increasing over the years. Risk factors for local recurrence and metastasis have been widely studied in highly sun-exposed areas of the body but less data exist about less sun-exposed ones. The main objective of this study is to compare the risk of local recurrence and metastases in patients with cSCC in highly sun-exposed areas compared to cSCC in less sun-exposed areas. Material and methods: A retrospective observational study was carried out, including 558 patients with histopathologically confirmed cSCC at the Reina Sofía University Hospital (HURS), Córdoba, during the period from 1 January 2017 to 31 December 2020. Demographic, clinical and anatomopathological data were collected. Results: Local recurrence occurs more often in highly sun-exposed areas (15.8%) compared to less sun-exposed ones (7.4%) (*p* < 0.05). However, no differences in the rate of metastases in both groups were found. The presence of affected surgical margins and tumor thickness were identified as independent risk factors for cSCC in low sun-exposure areas. Conclusions: cSCC located in anatomical areas of high sun exposure presented a greater risk of developing local recurrence in our population, which could suggest that these tumors have worse prognosis than those that lie in areas that do not receive chronic sun exposure.

## 1. Introduction

Cutaneous squamous cell carcinoma (cSCC) is a prevalent form of non-melanoma skin cancer (NMSC) arising from epidermal keratinocytes, with a lifetime incidence ranging between 7 and 11%. It is the second most common neoplasm within the NMSC group, accounting for approximately 20 to 25% of cases, trailing only behind basal cell carcinoma [1,2,3,4,5]. Over the last 30 years, the number of cases of cSCC has increased between 50% and 300% and in 2030 its incidence is projected to double in European countries. Whether this increase is real or corresponds to early detection of the disease is unknown [6]. 

The majority of cSCCs are successfully treated by surgical excision, achieving an overall 5-year survival rate of 90% after resection [5]. Nonetheless, there is a subset of cSCC with a more aggressive behavior, posing significant challenges in terms of recurrence and metastasis development and higher morbidity and mortality rates [7,8,9,10]. Around 3.7% to 5.2% of patients with cSCC have nodal metastasis, and 1.5% to 2.1% die of cSCC [11]. 

The majority of studies exploring the risk factors for local recurrence or metastasis in cSCC have primarily focused on tumors located in regions chronically exposed to ultraviolet radiation. Even the Eighth Edition of the AJCC Staging Manual excludes from its staging those cSCCs which arise outside of the head and neck areas [12]. It has been suggested that cSCCs located in minimally sun-exposed areas, such as the arms and legs, have a different mutational profile [13] and exhibit a more favorable prognosis than cSCCs occurring in the head and neck region [11,14,15,16,17,18]. Hence, this study was conducted to compare the disease progression of cSCC based on distinct anatomical sites with different quantities of sun exposure in terms of local recurrence and metastasis.

## 2. Materials and Methods

### 2.1. Study Population

A retrospective cohort study was conducted, including 558 patients with histopathologically confirmed cSCC treated at the Reina Sofía University Hospital (HURS), Córdoba, during the period from 1 January 2017 to 31 December 2020. 

Patients with cSCC in situ, keratoacanthomas, eruptive squamous atypia and basosquamous carcinomas were excluded. Patients with progeroid syndromes or genetic syndromes such as albinism, xeroderma pigmentosum, epidermolysis bullosa, Muir-Torre syndrome or Ferguson–Smith type, were also excluded due to an increased susceptibility to cutaneous epithelial neoplasm. 

### 2.2. Data Collection

Demographic, clinical and histopathological data of the cSCCS were extracted from dermatology and pathology departments of the HURS databases. The patient information collected included sex, age, type of community (rural or urban), Fitzpatrick skin phototype, immunosuppression, personal history of skin cancer (cutaneous cancer no melanoma (CCNM) or melanoma), phototherapy or radiotherapy. 

Tumor features included date of diagnosis, duration of evolution (according to patient or family information provided in history), location, size (cm), thickness (mm), Clark level, invasion of deep structures (fascia, muscle, perichondrium/periosteum, cartilage or bone), perineural or lymphovascular involvement, histologic differentiation, surgical margins’ status, distance from the edge to tumor, presence and date of local recurrence and lymph node or visceral metastasis. Clinically relevant perineural involvement was defined as tumor cells within the nerve sheath of a nerve lying deeper than the dermis or measuring ≥0.1 mm.

All cases were surgically removed through wide local excision, following the recommendations for surgical margins outlined in the European guidelines for cSCC treatment (excision with 6 to 10 mm peripheral clinical margins for high-risk to very-high-risk cSCC and 5 mm for low-risk cSCC) [19]. High-risk tumors were defined as those located on the head, neck, hands, feet, pretibia, or anogenital area, or those situated on the trunk and extremities with a preoperative clinical tumor diameter exceeding 2 cm [20]. Data regarding surgical reinterventions and adjuvant radiotherapy were also collected. All the patients included in the study were clinically monitored and, in those cases of high-risk cSCC, imaging tests were conducted following the recommendations of the Non-Melanoma Skin Cancer Multidisciplinary Tumor Board at HURS.

The head and neck and dorsum of the hands locations were considered highly sun-exposed areas and less sun-exposed areas were those located on the upper and lower limbs or trunk. Local recurrence (LR) was defined as the presence of biopsy-proven cSCC at the site of the previous excision. Metastasis was defined as the occurrence of biopsy-proven cSCC either in the lymph nodes that drain the affected area or in a distant organ.

The study was approved by the institutional review board of the Reina Sofía University Hospital (RSV001.1.0).

### 2.3. Statistical Analysis

A descriptive study was carried out calculating absolute and relative frequencies for the qualitative variables and arithmetic mean, standard deviation, and minimum and maximum value for the quantitative variables. The confidence interval (CI) at 95% security was estimated.

Parametric tests were used to compare the baseline characteristics between groups. To compare the quantitative variables, the Student *t*-test was used for independent groups and, for comparisons of more than 2 groups, the analysis of variance (ANOVA) was used. To compare the ordinal qualitative variables, the Mann–Whitney U-test was used for independent groups and the Kruskal–Wallis test for comparisons of more than 2 groups. Chi-square was used to compare the nominal qualitative variables.

Kaplan–Meier survival curves were used for estimating the risk of metastases and local recurrence for cSCCs according to their low/high degree of sun exposure. The Log-rank test was performed to assess the differences in survival between groups.

A secondary analysis was carried out to evaluate the risk factors for the development of local recurrence and metastasis in the group of patients with cSCC located in areas of low sun exposure, as well as the time from the moment of diagnosis of the primary cSCC and the appearance of local recurrence or metastasis. The period of risk for the development of a local recurrence or metastasis was calculated from the date of diagnosis of primary SCC to the date of last review, date of death or development of local recurrence or metastasis. A univariate analysis was performed to determine the variables influencing the prognosis in the subgroup of patients with cSCC in areas with low sun exposure.

All the contrasts were bilateral and those where *p* < 0.05 were considered significant. 

The data were collected, processed and analyzed with the statistical program SPSS v.17 (IBM, Armonk, NY, USA).

## 3. Results

A total of 558 patients with histopathologically confirmed cSCC treated at the HURS in Córdoba were included in the study. The demographic and clinical data of the patients are shown in Table 1 and their cSCC features are collected in Table 2.

Statistically significant differences (*p* < 0.05) were evident in terms of gender, with a marked male predominance (74.7%) observed in regions exposed to high levels of sunlight in contrast to those with low sun exposure. However, no other statistically significant differences were detected in relation to demographic data. The rural population exhibited a slight predominance in both groups. The distribution of Fitzpatrick skin phototypes was similar between both groups, with no individuals registered as having Fitzpatrick skin phototype IV. Regarding a history of previous skin cancer, cutaneous squamous cell carcinoma (CCNM) was more frequently diagnosed in the highly sun-exposed cSCC population, although these differences did not reach statistical significance.

All cases of highly sun-exposed cSCC were located on the head and neck, while less sun-exposed cSCCs were distributed across the upper limbs (50%), followed by the lower limbs (29.8%) and trunk (20.2%). The duration of cSCC evolution was similar in both groups (7.2 vs. 6 months). There were no statistically significant differences in the diameters or depth of the cSCC lesions between highly sun-exposed and less sun-exposed areas. Similarly, the degree of differentiation, perineural or lymphovascular involvement, invasion of deep tissues, and the presence of metastasis did not show statistically significant differences. A total of 6.9% of patients with tumors in highly sun-exposed areas received adjuvant radiotherapy, while in the group of tumors located in less sun-exposed areas, 6.3% did. None of the patients received adjuvant chemotherapy or immunotherapy.

However, notably, there were significant disparities in the rates of local recurrence, which were higher in highly sun-exposed areas (15.8%) compared to less sun-exposed areas (7.4%). Additionally, there was an increased rate of surgical margins affected by the lesion in the patient group with cSCC in highly sun-exposed regions. After the initial surgical intervention, the percentage of positive surgical margins was significantly higher in the cSCC group located on the head and neck (24.6%) compared to those in less sun-exposed areas (10.5%). The rate of surgical reinterventions was higher in patients with tumors located in highly sun-exposed areas, with 92 out of 114 tumors with positive margins requiring a second surgery (80.7%). The final rate of tumors achieving surgical margin clearance (R0) was slightly higher in the low sun-exposed group, although these intergroup differences did not reach statistical significance. No other statistically significant differences (*p* > 0.05) were observed in the comparison of the remaining variables.

The global estimated cumulative incidence rates for the development of local recurrence in patients with cSCC at 6, 12 and 24 months, were 6%, 9.3% and 12.1%, respectively (Table 3). In patients with a cSCC located in less sun-exposed areas the rates of developing local recurrence were lower at 6, 12 and 24 months (3.3%, 4.4% and 6.7%) compared with those cSCCs arising in highly sun-exposed areas (6.6%, 10.3% and 13.2%). The follow-up period was very similar in the highly and less sun-exposed groups (38 and 35 months, respectively). 

Statistically significant differences were detected when comparing survival curves for the incidence of local recurrence in patients stratified by their sun exposure levels (high sun exposure versus low sun exposure) (*p* = 0.036) (see Figure 1A). Nevertheless, no statistical significance was observed when assessing the development of metastasis (Figure 1B) or when analyzing both local recurrence and metastasis concurrently across both groups (Figure 1C).

The univariate analyses using the Cox regression model to explore potential prognostic factors in patients with cSCCs in less sun-exposed areas are succinctly summarized in Table 4. Significant predictors of the risk of local recurrence or metastasis in these patients on the univariate analysis included tumor thickness (RR, 1.1; CI, 1–1.2; *p* < 0.012) and the presence of positive surgical margins (RR, 2.5; CI, 1.2–5.1; *p* < 0.011). 

Other variables such as sex, age, phototype, previous skin cancer, immunosuppression, cSCC diameter, degree of differentiation or perineural invasion did not have a significant effect on the local recurrence or metastasis in univariate analysis.

## 4. Discussion

With the escalating incidence of cSCC, the accurate identification of risk factors for local recurrence and metastasis is of paramount importance in efforts to improve morbidity and mortality rates, as well as to implement sustainable healthcare policies [21]. While the most prominent risk factor for cSCC development is exposure to ultraviolet radiation (UV), encompassing PUVA therapy and tanning bed usage [22], the relationship between the quantity of sun exposure and local recurrence and metastasis remains relatively understudied.

Recent updates to the cSCC National Comprehensive Cancer Network (NCCN) guidelines underscore specific clinical and pathological characteristics as significant risk factors linked to heightened risks of recurrence, metastasis, and mortality in cSCC, yet they do not distinguish between high and low solar exposure [20]. Parameters such as tumor depth (measured in millimeters or anatomic depth) and perineural invasion exhibit a robust correlation with local recurrence. Furthermore, factors including tumor diameter (>2 cm), histological differentiation (particularly poor differentiation, desmoplastic, or acantholytic subtypes), lymphatic or vascular involvement, and immunosuppression, are recognized as contributing elements to recurrence, metastasis, and mortality [16,22,23]. A growth rate exceeding 4 mm per month along the tumor’s long axis is also associated with an unfavorable disease prognosis and an augmented risk of lymph node metastasis [6].

In our study cohort, which comprised 463 patients with cSCC located in highly sun-exposed areas and 95 patients with cSCC in less sun-exposed areas, the risk of local recurrence was notably higher in the highly sun-exposed group (15.8% vs. 7.4%) without differences in metastasis rates (4.1% vs. 5.3%). All cases of cSCC in highly sun-exposed areas were situated on the head and neck, while less sun-exposed areas included regions such as the trunk, upper extremities and lower extremities. Similar rates of recurrence have been described; cSCC cases localized to the head and neck region have rates of local recurrence and metastasis varying from 5% to 20%. In contrast, patients with cSCC on the extremities exhibit lower rates of local recurrence (3%) and metastasis (2.5%) [15,16].

It is important to acknowledge that the prognosis of patients with cSCC is significantly influenced by the tumor’s anatomical location. Certain regions, including the temporal area, auricular pavilion, and lip, have been associated with a higher incidence of metastasis [14]. However, data about specific risk factors for the local recurrence or metastasis of cSCCs located in low sun-exposed areas are scarce. In our study, we conducted an analysis to try and identify these risk factors in tumors located in less sun-exposed areas and determine if they allowed us to characterize the aggressiveness profile of these tumors in comparison to those located on the head and neck. Our data showed that initial surgical margin involvement and tumor depth were the characteristics associated with a shorter progression-free survival. 

Farah et al. conducted a study in which they analyzed histological prognostic factors in 230 patients with cSCC located in the head and neck versus other locations [24]. Regarding tumors located outside the head and neck, the authors demonstrated that the only independent risk factor for the development of metastasis was histological differentiation, while lymphovascular involvement was shown to predict worse disease-specific survival. Moreover, histological ulceration emerged as an independent risk factor for the development of metastasis in tumors located on the head and neck. In this study, no statistically significant differences were observed in the metastasis rate between both groups. On the other hand, in a recent systematic review and meta-analysis on prognostic factors in cSCC, the authors found that the tumor’s location on the head and neck did not exhibit statistically significant differences in nodal metastasis development [25]. These findings align with the results of our study and support the notion that tumors located in highly sun-exposed areas may not have a higher risk of nodal or distant metastasis in the absence of additional risk factors.

These findings raise the possibility that regions with lower levels of chronic sun exposure, such as the extremities, may be associated with a more favorable prognosis in the evolution of cSCC, while areas with higher solar exposure carry an elevated risk of developing a poorer long-term prognosis [17,18,26,27,28,29,30,31,32,33]. There is a growing focus on understanding the role of UV radiation in skin carcinogenesis, seeking to establish the link between sun exposure and carcinogenesis [34]. Robust evidence underscores the impact of UV radiation on skin cancer development, as it promotes mutations in cellular DNA, induces a state of relative cutaneous immunosuppression, and may facilitate persistent infection with human papillomaviruses. Furthermore, genes involved in DNA repair, such as p53, are potential targets of UV radiation. Mutations in the tumor suppressor gene p53 have been reported in 50% to 90% of cSCC cases [35]. Recently, the fusion gene EGFR-PPARGC1A has emerged as a possible causative agent in cSCC. Both mutations had been predominantly discovered in chronically UV-exposed areas, suggesting an association with chronic sun exposure [33,34]. Additionally, higher PD-L1 expression levels have also been correlated with cSCC in sun-exposed skin [15].

In accordance with the existing literature, which typically reports a male-to-female ratio of approximately 3:1 [36], we noted a notable male predominance (74.7%) in highly sun-exposed areas, likely caused by male pattern hair loss and occupational exposure [37]. However, this gender bias was not observed in less sun-exposed areas. In this latter group, the incidence was marginally higher in women (56.8%) than in men (43.2%), revealing statistically significant differences with a *p*-value of <0.001. This intriguing observation could potentially be attributed to the extended lifespan of women, which leads to a more prolonged potential duration of UV radiation exposure.

In addition, our study revealed significant disparities in the occurrence of surgical margin involvement among patients with cSCC, which is known to be a risk factor for local recurrence and metastasis. A substantially higher rate of initial surgical margin involvement was noted in individuals with SCC localized in regions with high sun exposure. This discrepancy may be ascribed to the intricate anatomy of facial areas, where the presence of vital functional structures frequently constrains the feasibility of extensive excisions, consequently elevating the risk of compromised margins. This higher rate of initially affected surgical margins resulted in a greater number of surgical reinterventions in the group of tumors located in highly sun-exposed areas. The final clearance of surgical margins was very similar in both groups, which does not account for the differences in local recurrence rates between the two groups. To mitigate this risk, Mohs surgery represents a viable treatment option for high-risk cSCC cases. This technique has demonstrated superior outcomes in this specific subgroup of patients [15]. The fact that all our patients were treated with conventional surgery could explain the higher rate of affected margins obtained in our study, especially in tumors located on the head and neck, compared to studies where Mohs surgery is performed for cSCC. 

Another aspect that could potentially impact the prognosis of less sun-exposed tumors is delayed diagnosis due to their location in less visible areas such as the trunk or the limbs. This could lead to a potential delay in surgery, which, in turn, might influence the choice of surgical approach in cases of larger or deeper tumors, as well as the possibility of achieving an R0 resection. However, in our sample, we did not observe differences in the time of progression between tumors located in highly and less sun-exposed areas, which would contradict the possibility that a diagnostic delay had influenced the prognosis of either group.

Apart from the intrinsic limitations of a retrospective study, this study is subject to certain additional limitations. As a referral center, our institution often receives patients with cSCC, who have a poor prognosis, from other healthcare facilities. This scenario could potentially lead to an overestimation of the risk of local recurrence and metastasis in our study population, as it may include individuals with more advanced or severe cases of cSCC. Furthermore, it is important to note that some patients received treatment from other surgical teams that chose clinical observation as the course of action after identifying affected surgical margins, rather than opting for reintervention. This variability in treatment approaches adds complexity to our study. In addition, in several cases of tumors requiring surgical reintervention due to affected surgical margins, no tumor was found in the surgical biopsy. Consequently, the possibility of some false positives could not be completely ruled out, which may partially explain the increased risk of local recurrence in patients unable to achieve an R0 status in the initial surgery. Finally, the relatively small sample size in the group of patients with less sun-exposed cSCC could have underestimated our univariate analyses results. Moreover, this fact constrains the statistical power available for conducting multivariate analyses, given the numerous potential risk factors at play.

Future research endeavors should aim to address these limitations by utilizing larger and more diverse cohorts to identify and analyze key prognostic factors. An enhanced understanding of the intricate relationship between cSCC in regions with low solar exposure and its prognostic implications will significantly contribute to the enhancement of its clinical management and the formulation of preventive strategies for this particular subset of patients.

## 5. Conclusions

The anatomical location of cSCC in areas of high sun exposure seems to be associated with an increased risk of local recurrence in our study population, hinting at a potentially less favorable prognosis compared to tumors located in less sun-exposed regions. These differences could be explained because some risk factors for the local recurrence of cSCC appear more frequently on highly sun-exposed skin (surgical margin involvement or deep invasion). Highly sun-exposed areas may exhibit distinct genetic backgrounds that render them more aggressive and prone to local recurrence and metastasis.

A better understanding the oncogenic characteristics related to cSCC in regions of high and low sun exposure will improve the management and therapeutic strategies for these patients.

## Figures and Tables

**Figure 1 cancers-15-05037-f001:**
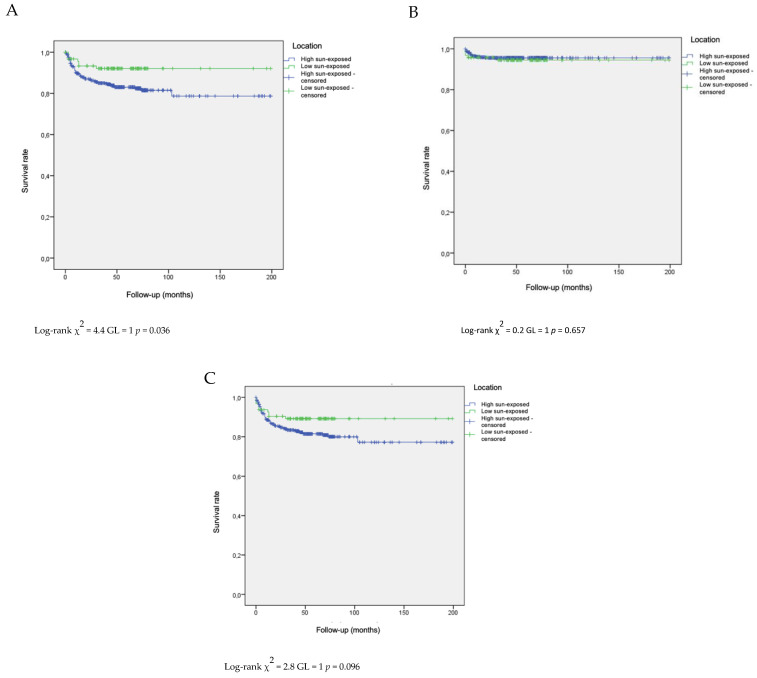
Survival curve for the development of local recurrence (**A**), of metastasis (**B**) and local recurrence or metastasis (**C**).

**Table 1 cancers-15-05037-t001:** Demographic and clinical data.

Characteristics of Patients	cSCC in Highly Sun-Exposed Areas	cSCC in Less Sun-Exposed Areas	*p* Value
	N (%) o Mean (DS)	N (%) o Mean (DS)	
No. of patients			
	463	95	-
Sex			
Male	346 (74.7%)	41 (43.2%)	<0.001
Female	117 (25.3%)	54 (56.8%)	
Age (years)			
	78.7 (10.1)	76.9 (12.8)	NS
Type of community			
Urban	198 (43%)	40 (42.1%)	NS
Rural	263 (57%)	55 (57.9%)	
Fitzpatrick skin phototype			
I	16 (18.8%)	1 (6.7%)	
II	35 (41.2%)	7 (46.7%)	NS
III	34 (40%)	7 (46.7%)	
Previous skin cancer			
No	207 (44.7%)	57 (60%)	
CCNM	249 (53.8%)	36 (37.9%)	NS
Melanoma	5 (1.1%)	1 (1.1%)	
CCNM + melanoma	2 (0.4%)	1 (1.1%)	
Immunosuppression			
No	390 (84.2%)	75 (78.9%)	
Pharmacological	60 (13%)	14 (14.7%)	NS
No-pharmacological immunosuppression	13 (2.8%)	6 (6.3%)	

NS: non-significative

**Table 2 cancers-15-05037-t002:** cSCC features of the patients in the study.

SCC Characteristics	cSCC in Highly Sun-Exposed Areas	cSCC in Less Sun-Exposed Areas	*p* Value
	N (%) o Mean (DS)	N (%) o Mean (DS)	
SCC localization			
Head and neck	463 (100%)	-	
Upper limbs	-	47 (50%)	
Lower limbs	-	28 (29.8%)	
Trunk	-	19 (20.2%)	
High-risk tumors			
	463 (100%)	65 (68.7%)	
Duration of evolution (months)			
	7.2 (8.3)	6 (6.7)	NS
SCC major diameter (cm)			
	1.7 (1.2)	2 (1.6)	NS
SCC minor diameter (cm)			
	1.4 (1)	1.7 (1.4)	NS
Depth (mm)			
	5.7 (4.7)	5.8 (4.7)	NS
Degree of differentiation			
Poor differentiated	69 (15.1%)	5 (5.6%)	
Moderate differentiated	201 (43.9%)	42 (46.7%)	NS
Well differentiated	188 (41%)	43 (47.8%)	
Diagnosis of PNI			
No	423 (91.3%)	87 (95.6%)	
Incidental	23 (4.9%)	2 (2.2%)	NS
Clinically relevant	17 (3.7%)	2 (2.2%)	
Lymphovascular involvement			
No	444 (97.6%)	95 (100%)	NS
Yes	11 (2.4%)	-	
Deep of invasion			
No	390 (86.1%)	80 (86%)	
Fascia	47 (10.4%)	11 (11.8%)	
Muscle	10 (2.2%)	2 (2.2%)	NS
Cartilage	3 (0.7%)	-	
Bone	3 (0.7%)	-	
Metastasis			
No	444 (95.9%)	90 (94.7%)	NS
Yes	19 (4.1%)	5 (5.3%)	
Local recurrence			
No	390 (84.2%)	88 (92.6%)	0.03
Yes	73 (15.8%)	7 (7.4%)	
Surgical margins			
Negative	340 (74.9%)	82 (89.1%)	0.002
Positive	114 (24.6%)	10 (10.5%)	
Lateral margin positive	59 (12.7%)	4 (4.2%)	
Deep margin positive	55 (11.8%)	6 (6.3%)	
Surgical reintervention after positive margins			
No	22/114 (19.3%)	5/10 (50%)	0.024
Yes	92/114 (80.7%)	5/10 (50%)	
Final R0 status			
	394 (85.1%)	86 (90.5%)	NS
Adjuvant radiotherapy			
No	431 (93.1%)	89 (93.7%)	NS
Yes	32 (6.9%)	6 (6.3%)	
Follow-up (months)	38 (5.7)	35 (6.4)	NS

PNI: perineural involvement. R0: no residual tumor in the histopathological report. NS: non-significative.

**Table 3 cancers-15-05037-t003:** Estimated cumulative probability of developing metastasis or local recurrence in patients with cSCC.

	6 Months	12 Months	24 Months
Local recurrence or metastasis High sun exposure Low sun exposure	7.5%	10.8%	13.9%
	7.8%	11.5%	14.8%
	6.3%	7.4%	9.6%
Local recurrence High sun exposure Low sun exposure	6%	9.3%	12.1%
	6.6%	10.3%	13.2%
	3.3%	4.4%	6.7%
Metastasis High sun exposure Low sun exposure	3.3%	3.9%	4.4%
	3.2%	3.9%	4.4%
	4.2%	4.2%	4.2%

**Table 4 cancers-15-05037-t004:** Results of univariate analyses of low sun exposure areas using the Cox regression model.

Variables	RR	CI 95%	*p* Value
Sex	0.5	(0.1–1.7)	NS
Age (years)	1	(1–1.1)	NS
Phototype	4.1	(0.5–36)	NS
Previous skin cancer	1.3	(0.5–3.5)	NS
Immunosuppression	0.8	(0.2–2.7)	NS
SCC larger diameter (cm)	1.2	(0.9–1.6)	NS
SCC minor diameter (cm)	1.2	(0.9–1.7)	NS
SCC depth (mm)	1.1	(1–1.2)	0.012
Degree of differentiation	0.4	(0.1–1)	NS
Perineural invasion	2.8	(0.3–22.4)	NS
Surgical margins	2.5	(1.2–5.1)	0.011

NS: non-significative.

## Data Availability

Supporting data cand be provided by contacting with the corresponding author.

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
