# Peer review of "Increased Risk of Local Recurrence in Cutaneous Squamous Cell Carcinoma Arising in Sun-Exposed Skin: A Retrospective Cohort Study"

_cancers, 2023, doi:10.3390/cancers15205037_

Round 1
Reviewer 1 Report
The purpose of the manuscript is to compare the risk in patients with cutaneous squamous cell carcinoma on high exposed areas compared to carcinomas on low exposed areas.
In my opinion, the following points need to be better clarified:
-the authors should add information about the type (and duration) of follow up;
-the authors should provide information on how they evaluated the duration of carcinoma evolution, according to the location of carcinoma;
-the authors should specify the reason why the univariate analysis was not performed for high exposed areas carcinomas;
-the difference between recurrence and metastasis should be better discussed, according to the literature;
-margin status and thickness are influenced by location of carcinoma, that influenced the time of diagnosis and the type of surgery. Thus, risk of recurrence is influenced by location of tumor. On the other hand, risk of metastasis is also influenced by the biology of tumor: additional histological characteristics of squamous cell carcinomas (mitosis, histological variants, grading) and information about molecular factors should be discussed.
Author Response
REVIEWER 1
The purpose of the manuscript is to compare the risk in patients with cutaneous squamous cell carcinoma on high exposed areas compared to carcinomas on low exposed areas.
In my opinion, the following points need to be better clarified:
Dear reviewer, thank you very much, we really appreciate your helpful remarks.
-the authors should add information about the type (and duration) of follow up;
The patients were clinically monitored following the current protocols and guidelines of the NCCN and EADV. In cases of high-risk cSCC, imaging tests were conducted after an individual assessment of each case, following the recommendations of the Non-Melanoma Skin Cancer Multidisciplinary Tumor Board at Reina Sofía University Hospital. The duration of the follow-up has been included in the results section, and we have updated the methods section with a note regarding the type of follow-up.
-the authors should provide information on how they evaluated the duration of carcinoma evolution, according to the location of carcinoma;
This information was collected from the patient's medical record, where family members and patients are systematically asked about the date they first noticed the tumor's initial manifestation. We have included a comment in the methods section.
-the authors should specify the reason why the univariate analysis was not performed for high exposed areas carcinomas;
Since the risk factors for the development of local recurrence or metastasis in patients with high-exposed areas cSCC have been extensively studied in the literature, we preferred to focus on the subgroup of tumors that appear in low sun-exposed areas to determine if their risk factors might differ from those established for cSCC located in high sun-exposed areas. We have provided this justification in the text.
-the difference between recurrence and metastasis should be better discussed, according to the literature;
We have expanded the discussion about the differences between local recurrence and metastasis as suggested by the reviewer, in line with articles from the literature.
-margin status and thickness are influenced by location of carcinoma, that influenced the time of diagnosis and the type of surgery. Thus, risk of recurrence is influenced by location of tumor. On the other hand, risk of metastasis is also influenced by the biology of tumor: additional histological characteristics of squamous cell carcinomas (mitosis, histological variants, grading) and information about molecular factors should be discussed.
Thank you very much for your comments. We fully agree with the reviewer that margin status is a critically important variable regarding the possibility of developing subsequent local recurrence. We have added a paragraph in the text referencing this point as highlighted by the reviewer. Additionally, we have included a reference in the article regarding the histological risk factors for the development of metastasis (Farah et al. JEADV 2022).
Reviewer 2 Report
I think the cohort of cases is sound. I have the following suggestions.
Given that the conclusion states that SCC in sun exposed sites is a risk factor for local recurrence and these mostly occur on the head and neck where anatomic constraints limit margin clearance, so Mohs is more frequently done, is it then true that these margins are almost always clear because Mohs is performed, so margin status plays very little role in these cases? How many of these cases were Mohs versus surgical excision?
The lack of correlation with perineural invasion I think needs to be further addressed since it is a departure from prior studies. Was the caliber of the nerves assessed in this study and if so how many were 0.1 mm or greater?
There is no definition of recurrence (e.g. local, etc.). What does this mean? Also, head and neck cancers frequently spread to the parotid. Is this considered distant? A definition of local versus locoregional spread is necessary here.
How many patients received radiation as a form of therapy in this study? I don't see that that was accounted for here. If none did then that should be stated.
How many patients received adjuvant chemotherapy if any? This needs to be addressed also.
Author Response
REVIEWER 2
I think the cohort of cases is sound. I have the following suggestions.
Dear reviewer, thank you very much, we really appreciate your helpful remarks.
Given that the conclusion states that SCC in sun exposed sites is a risk factor for local recurrence and these mostly occur on the head and neck where anatomic constraints limit margin clearance, so Mohs is more frequently done, is it then true that these margins are almost always clear because Mohs is performed, so margin status plays very little role in these cases? How many of these cases were Mohs versus surgical excision?
Thank you very much for your comment. In our center, Mohs surgery is not available, so all cases were surgically removed through wide local excision, following the recommendations for surgical margins outlined in the European guidelines for cSCC treatment (excision with 6 to 10 mm peripheral clinical margins for high-risk to very-high-risk cSCC and 4 mm for low-risk cSCC). This could explain the presence of higher local recurrence rates than in articles where Mohs surgery has been used for tumor treatment. We have included this information in the text.
The lack of correlation with perineural invasion I think needs to be further addressed since it is a departure from prior studies. Was the caliber of the nerves assessed in this study and if so how many were 0.1 mm or greater?
Thank you for your comment. The assessment of nerve caliber has been included in the results section and Table 2.
There is no definition of recurrence (e.g. local, etc.). What does this mean? Also, head and neck cancers frequently spread to the parotid. Is this considered distant? A definition of local versus locoregional spread is necessary here.
We have included the specific definition of local recurrence and metastasis in the methods section.
How many patients received radiation as a form of therapy in this study? I don't see that that was accounted for here. If none did then that should be stated.
How many patients received adjuvant chemotherapy if any? This needs to be addressed also.
We have included in the text the frequency of adjuvant radiotherapy or chemotherapy in our sample.
Round 2
Reviewer 2 Report
I have no further comments.
I have no further comments.
Author Response
Thank you very much for responding to the comments of the reviewers. Taken together, I ask to highlight one issue: The initial excision was performed with safety margins (and not Mohs surgery). Safety margins are often not possible due to anatomic reasons in the head/neck area, but at the trunk/extremities. Thus, lower safety margins in the head/neck tumors could explain the higher rate of margin positive tumors here and the higher rate of local recurrences as compared to tumors on trunk/extremities. Can the authors discuss that and give numbers on how many patients were not operable with the projected safety margins (4mm in low risk cSCC and 6-10mm in high risk cSCC) in the sun exposed versus the non-sun exposed groups? Thank you.
We fully understand the Editor's concerns and would like to address this issue. When we refer to tumors located in areas of low sun exposure, we are specifically talking about those situated on the trunk and extremities. However, this does not necessarily equate to low-risk tumors. In our study, tumors located on the trunk or extremities with a major diameter exceeding 2 cm, as well as those on the feet, hands, or pretibial areas, are considered high-risk tumors. Consequently, they are treated with the same safety margins as tumors located on the face and neck, all of which are categorized as high-risk tumors.
In our sample, 65 (68.7%) tumors in the low-sun exposed group can be considered high-risk due to their location or size exceeding 2 cm. Hence, the differences between both groups concerning the choice of preoperative surgical margins are less significant than one might initially assume.
Additionally, the presence of affected margins in the primary tumor excision does not necessarily imply that no further surgical treatment will be performed. The percentages of margin involvement in the initial surgical specimens were presented in Table 2. However, in most cases where clear margins cannot be achieved, if the patient is a candidate for surgery, a new intervention is typically chosen to ensure an R0 surgery. We have reanalyzed our sample to assess the number of reoperations conducted in each group after affected margins, and we have added this data to Table 2. The final rate of R0 status were very similar in both groups and showed no statistical differences.
We have also updated the discussion section with the new data provided and the limitations section.